# A Perspective of Molecular Cytogenomics, Toxicology, and Epigenetics for the Increase of Heterochromatic Regions and Retrotransposable Elements in Tambaqui (*Colossoma macropomum*) Exposed to the Parasiticide Trichlorfon

**DOI:** 10.3390/ani12151945

**Published:** 2022-07-31

**Authors:** Maria dos Santos Costa, Hallana Cristina Menezes da Silva, Simone Cardoso Soares, Ramon Marin Favarato, Eliana Feldberg, Ana Lúcia Silva Gomes, Roberto Ferreira Artoni, Daniele Aparecida Matoso

**Affiliations:** 1Programa de Pós-Graduação em Genética, Conservação e Biologia Evolutiva, Instituto Nacional de Pesquisas da Amazônia (Amazonas), Manaus 69067-375, Brazil; marrycuesta@gmail.com (M.d.S.C.); saybio@hotmail.com (S.C.S.); 2Departamento de Odontologia, Universidade do Estado do Amazonas, Manaus 69065-001, Brazil; 3Laboratório de Genética Animal, Instituto Nacional de Pesquisas da Amazônia, Manaus 69067-375, Brazil; ramonfavarato@gmail.com (R.M.F.); feldberg@inpa.gov.br (E.F.); 4Laboratório de Parasitologia de Animais Aquáticos, Departamento de Parasitologia, Instituto de Ciências Biológicas, Universidade Federal do Amazonas, Manaus 69067-005, Brazil; anapaimagomes@gmail.com; 5Laboratório de Genética e Evolução, Departamento de Biologia Estrutural, Molecular e Genética, Universidade Estadual de Ponta Grossa, Ponta Grossa 84030-900, Brazil; rfartoni@gmail.com; 6Laboratório de Biotecnologia e Citogenômica Animal, Departamento de Genética, Instituto de Ciências Biológicas, Universidade Federal do Amazonas, Manaus 69067-005, Brazil; danielematoso@yahoo.com.br

**Keywords:** FISH, heterochromatinization, organophosphate, repetitive elements

## Abstract

**Simple Summary:**

The aim of the present study was to evaluate the Trichlorfon effects on the retrotransposable elements in tambaqui (*Colossoma macropomum*) genome, which is a highly popular and well-known fish in the Amazon with a large reproduction number mediated by pisciculture. Thereby, tambaqui specimens were submitted to two different Trichlorfon concentrations (30% and 50% of LC_50–96 h_) under experimental conditions. The retrotransposons were analyzed using the FISH technique and the heterochromatin standard with the C-band technique. The retrotransposons studied presented a dispersed distribution profile in the tambaqui karyotype with *Rex*3 being more prominent than the others, showing the greatest increase in markings. Furthermore, the heterochromatin profile showed that these retrotransposons can be found in the heterochromatic portions of the chromosomes. Thus, it was observed that Trichlorfon has an activation mechanism for these retroelements, especially *Rex*3.

**Abstract:**

*Rex* retroelements are the best-known transposable elements class and are broadly distributed through fish and also individual genomes, playing an important role in their evolutionary dynamics. Several agents can stress these elements; among them, there are some parasitic compounds such as the organochlorophosphate Trichlorfon. Consequently, knowing that the organochlorophosphate Trichlorfon is indiscriminately used as an antiparasitic in aquaculture, the current study aimed to analyze the effects of this compound on the activation of the Transposable Elements (TEs) *Rex*1, *Rex*3, and *Rex*6 and the structure of heterochromatin in the mitotic chromosomes of the tambaqui (*Colossoma macropomum*). For this, two concentrations of the pesticide were used: 30% (0.261 mg/L) and 50% (0.435 mg/L) of the recommended LC_50–96 h_ concentration (0.87 mg/L) for this fish species. The results revealed a dispersed distribution for *Rex*1 and *Rex*6 retroelements. *Rex*3 showed an increase in both marking intensity and distribution, as well as enhanced chromosomal heterochromatinization. This probably happened by the mediation of epigenetic adaptive mechanisms, causing the retroelement mobilization to be repressed. However, this behavior was most evident when Trichlorfon concentrations and exposure times were the greatest, reflecting the genetic flexibility necessary for this species to successfully adapt to environmental changes.

## 1. Introduction

The organochlorophosphate Trichlorfon (dimethyl [2,2,2-trichloro-1-hydroxyethyl] phosphonate), commercially known as Dipterex 500^®^, Tugon, Masoten^®^ or Neguvon^®^, is widely used against aquacultural parasites [1]. High concentrations of this compound are able to inhibit the acetylcholinesterase enzyme (AChE), causing the accumulation of acetylcholine (ACh) in the nerve synapses, inhibiting its proper function [2,3,4]. In the neuronal endplates, sublethal doses of Trichlorfon can cause significant physiological, behavioral, and ecological damages [5].

Organochlorophosphate is a class of pesticides commonly used in commercial pisciculture [6] because it remains in the environment for a relatively short period of time, being considered less harmful to the environment [7,8]. Brazilian environmental legislation [9] establishes a maximum concentration of 1.0 mg/L for organochlorophosphates in freshwater, however, it is known that aquaculturists are used to applying a huge range of doses. There are reports of some aquaculturists using therapeutic Trichlorfon baths with concentrations ranging from 0.13 to 25 g/L [7]. The uncontrolled use of Trichlorfon in pisciculture can cause the hydrolyzation of this compound, resulting in the formation of dichlorvos, which is a metabolite associated with high levels of toxicity [10], as shown by the studies of Veiga et al., 2002, Sinha et al., 2010, and Venturini et al., 2014 [11,12,13].

The city of Rio Preto da Eva, in Amazonas, northern Brazil, is the second-largest producer of tambaqui (*Colossoma macropomum*) [Cuvier, 1818] in the country [14]. This fish species can reach about 108 cm in length and weigh up to 40 kg [15]. *C. macropomum* production statistics for Brazil show a significant output increase in recent years; with a total 691,700 tons of national cultivated fish production in 2018, *C. macropomum* production represented 43.7% of the total volume (302,235 tons).

Studies of the “genomic health” of the target species are of interest to aquaculture, which is a highly productive sector, being key to establishing management and conservation strategies for the preservation of the genetic heritage of this species. In this context, the genomes of teleosts, the taxonomic group to which *C. macropomum* belongs, have been shown to contain extensive volumes of repetitive DNA [16], including Transposable Elements (TEs), which consist of DNA sequences with the capacity to change their chromosomal location within the same genome [17,18].

*Rex* elements are the most studied TE class for Brazilian fish and are widely distributed in teleosts [19,20]. The retrotransposons *Rex*1, *Rex*3, and *Rex*6, initially described for teleosts from the *Xiphophorus* genome by Volff et al., 1999, 2000, 2001 [21,22,23], have an important role in fish evolution dynamics, and are widely distributed within teleost genomes, where a great variety of distributional patterns exists [24,25,26,27]. These repeated elements represent a substantial fraction of vertebrate genomes, contributing significantly to the evolution of this species [28,29,30].

Fluorescent In Situ Hybridization (FISH) is an extensively used technique to map DNA sequences of interest, such as the *Rex* elements [23,24,25,26,27]. Similarly, classical studies using the C-banding technique, which detects heterochromatic regions in the genome, are very useful to verify the presence of natural polymorphisms in several groups of organisms [31]. Heterochromatin is a condensed and transcriptionally silent region of the genome and it is preferably assembled in regions that harbor repetitive elements, such as satellite DNA and TEs [29,32,33].

Studies related to the activity of these retroelements to environmental adaptation make it clear that they play a fundamental role in phenotypic adaptation [34,35]. The main purpose of this study was to test the effects of the parasiticide Trichlorfon on the transposition of *Rex* retrotransposable elements and their distribution in the heterochromatin of the mitotic chromosomes of *C. macropomum*.

## 2. Materials and Methods

### 2.1. Experimental Design: Acquisition, Acclimatization, Tissue Collection, and Chromosomal Preparations

Individual *C. macropomum* juveniles were selected from a stock located at Universidade Federal do Amazonas (UFAM) Experimental Farm, on BR-174, kilometer 38, near Manaus, Amazonas, Brazil (2°38′57.6″ S 60°3′11″ W). The selected fish were transferred to the Wet Laboratory for Aquatic Animal Parasitology, Physiology, and Genetics and were acclimatized in polyethylene boxes at a temperature of 25 °C (±2 °C) for 40 days, monitoring the vital signs of the animals all the time. At first, the fish were fed twice a day with commercial feed containing 40% of protein and then kept unfed for 24 h before the beginning of the experiment, in order to guarantee the gastrointestinal canal emptiness.

This study adopted the same LC_50–96 h_ exposure methodology applied by Silva et al., 2019 and Duncan et al., 2019 [3,4]. Three groups were evaluated, one group for the control and two of them with sublethal concentrations: concentration 0 (C0) corresponding to the control group (with no Trichlorfon added to the water), concentration 1 (C1) with 30% LC_50–96 h_ (corresponding to 0.261 mg/L of Trichlorfon added to the water tank), and concentration 2 (C2) with 50% LC_50–96 h_ (corresponding to 0.435 mg/L of Trichlorfon added). The following physicochemical water variables were evaluated: temperature, pH, and dissolved oxygen, using a Hanna multiprobe (model HI98194). To reduce the production of large volumes of contaminated water, the experiments were conducted with a static system that did not involve water removal. Nine polyethylene tanks with 60 L capacity were used, with six fish in each. The tanks for each treatment were randomly chosen, as shown in Figure 1. The fish were also randomly collected after 48, 72, and 96 h, in each experimental group, with their respective control, totaling 54 samples (all individuals were kept alive until the end of the experiment). Then, 18 specimens per treatment were analyzed for each concentration (C0, C1, C2). Proceeding the collection, fish were transferred to the Biotechnology and Animal Cytogenomics Laboratory at UFAM where they were measured (7.3 ± 0.9 cm), weighed (13.6 ± 4.3 g), and then submitted to cytogenetic analysis.

The methodology described by Gold et al., 1990 [36], with some modifications, was used to obtain the chromosomal preparations. The kidney was removed and transferred to a glass tank containing 4 mL of RPMI 1640 culture medium at room temperature. Then, 4 μL of colchicin 0.0125% were added to the dissociated material, still at room temperature, for 30 min. After this, the material was transferred to a 10 mL Falcon tube and centrifuged for 10 min at 900 rpm. The supernatant was removed with a Pasteur pipette and then was added to a tube with a hypotonic 0.075M KCl solution. The material was resuspended and incubated for 30 min, now at 37 °C. The slides were submitted to the C-banding technique, according to Sumner 1972. The relation between the chromosomes’ arms followed the one described by Levan et al., 1964 [37].

### 2.2. Extraction of Total DNA, PCR, and Fluorescence In Situ Hybridization (FISH)

For total DNA extraction, the Wizard^®^ HMW DNA Extraction Kit (Promega, Madinson, WI, USA) was used, following the manufacturer’s instructions, with the DNA quantified by a NanoVue Plus spectrophotometer (GE Healthcare, Chicago, IL, USA). Amplification of the *Rex* retrotransposable elements used an Applied Biosystems thermocycler, with probes obtained through a Polymerase Chain Reaction (PCR), using the primers: *Rex*1 RTX1-F1 (5′TTC TCC AGT GCC TTC AAC ACC) and RTX1-R3 (5′TCC CTC AGC AGA AAG AGT CTG CTC) [22], *Rex*3 RTX3-F3 (5′-CGG TGA YAA AGG GCA GCC CTG-3′) and RTX3-R3 (5′-TGG CAG ACN GGG GTG GTG GT-3′) [21,22], and *Rex*6-Medf1 (5′-TAA AGC ATA CAT GGA GCG CCA C-3′), and *Rex*6-Medr2 (5′-GGT CCT CTA CCA GAG GCC TGG G-3′) [23]. Reaction effectiveness was analyzed by running a 1% agarose gel with samples quantified by a NanoVue Plus spectrophotometer (GE Healthcare, Chicago, IL, USA).

The PCR products (*Rex*1, *Rex*3 and *Rex*6) were marked with Digoxigenin via Nick Translation reaction, using a DIG-Nick^TM^ Translation Mix kit (Roche, Basel, Switzerland), following the manufacturer’s instructions to obtain the probes. For probe signal detection, anti-rhodamine-digoxigenin (Roche) antibody was used. Fluorescent In Situ Hybridization (FISH) was deployed with 77% restriction as described by Pinkel et al., 1986 [34], but with some modifications: 2.5 ng/μL DNA, 25 μL of 50% deionized formamide, 10 μL 50% dextran sulfate, and 5 μL of 20xSSC at 37 °C for 18 h. Post-hybridization washings were made with 15% formamide and 2 × SSC/Tween at 0.05%. Chromosomes were stained with DAPI (2 mg/mL) using a Vectashield^®^ (Vector) mounting support. After analysis and counts of metaphasic mitotic chromosomes, about 60 metaphases per experimental condition and 18 specimens per treatment (obtained using C-Banding and FISH) were photographed using fluorescence microscopy with an Olympus BX51 and an immersion objective lens to capture images, plus Image-PRO MC 6.0 software (Media Cybernetics, Rockville, MD 20852, USA). In each case, the best metaphase examples (those with good chromosomal pairing) were used. To set up the karyotypes, the Adobe Photoshop v. 23.4.1 software (Adobe Systems, San Jose, CA, USA) was used and the chromosomes were recorded, paired, and grouped according to their morphology in decreasing size order [35]. Chromosomes were measured using the Image Processing and Analysis in Java (Image J, Bethesda, MA, USA).

### 2.3. Ethical Statement

The care and use of experimental animals complied with the Ethics Committee on Animal Experimentation—CEUA/UFAM, animal welfare laws, guidelines, and policies as approved by license number 030/2018.

## 3. Results

Analysis at metaphase confirmed 2 n = 54 chromosomes (26 m + 28 sm) and the fundamental number (FN) as 108, as already established for *C. macropomum* by Barbosa et al., 2014 and Ribeiro et al., 2014 [28,38]. Neither intra nor interindividual polymorphisms were observed in the control group (C0 = 0.0 mg/L of Trichlorfon). The number of metaphases per slide were significantly high with the increase of the Trichlorfon concentration and the exposure time of the animals to the stress agent, exceeding 100 countable metaphases per slide at a concentration of 50% LC_50–96 h_ (0.435 mg/L) for 96 h.

For constitutive heterochromatin distribution, it was observed that most heterochromatic blocks were concentrated in the centromeric portions of all chromosomes from the control group (Figure 2a-I) while individuals from the experimental group that were exposed to 30% LC_50–96 h_ (0.261 mg/L) for 48 h (Figure 2a-II) showed remarkable blocks in the terminal regions of the selected chromosomal pairs 7, 8, 16, and 18.

Those individuals exposed to 50% LC_50–96 h_-48 h (Figure 2a-III), also showed remarkable blocks in the centromeric regions and bi-telomeric patterning on pair 4, in addition to heterochromatic blocks in the terminal regions of the pairs 7, 8, and 16 and interstitial blocks on pair 18. Individuals exposed to 50% LC_50–96 h_-72 h (Figure 2a-IV) showed remarkable chromosomal markings in the centromeric regions, bi-telomeric markings on pairs 4 and 8, heterochromatic blocks in the terminal regions of pairs 7 and 16, and interstitial blocks on pair 18. Individuals exposed to the highest Trichlorfon concentration, that is 50% LC_50–96 h_-96 h (Figure 2a-V), in addition to heterochromatic markings in the centromeric regions, showed heterochromatic blocks much more remarkable than those seen in individuals from all other tested experimental treatments, as well as the presence of centromeric and terminal blocks on pair 1, bi-telomeric markings on pairs 4, 7, and 8, markings in the terminal region of pair 16, interstitial blocks on pair 18, and terminal markings on pair 26.

For *Rex*3, the individuals from the control group showed diffuse and poorly defined markings (Figure 2b-VII). After 48 h exposure in the concentration of 30% LC_50–96 h_, there was an increase in the markings for this retroelement at telomeric portions (Figure 2b-VIII). However, after 72 h (Figure 2b-IX), an evident increase in reactivity at *Rex*3 sites was visible, with some instances including the terminal regions of some chromosome pairs. This same pattern occurred at the concentration of 50% LC_50–96 h_ for 48 h exposure (Figure 2b-X). With a notable increase in sites for this retrotransposon at 50% LC_50–96 h_ concentration for 72 h, with remarkable markings throughout the chromosome set, showing well-defined blocks in the terminal portions of all chromosomes (both arms), possibly due to the association of *Rex*3 sites with the formation of heterochromatic blocks (Figure 2b-XI). At the 50% concentration of LC_50–96 h_ for 96 h (Figure 2b-XII), there is an evident increase in the distribution and intensity of the number of *Rex*3 sites, with a large increase in markings. The surface of almost all chromosomes was marked along the telomeric and centromeric regions, including both heterochromatic and euchromatic portions. The heterochromatin corresponds to an important fraction of the genome and constitute large blocks near centromeres and telomeres chromosomes [39]. The enrichment of TES in heterochromatin regions has been now considered a general aspect of this fraction [40]. In this sense, the increase in *Rex*3 retroelements after treatment also correlates with the increase in heterochromatin. However, this increase in *Rex*3 was not fully accompanied by the increase in heterochromatin because the experiment was performed for 96 h and there was no time for the heterochromatinization process to occur.

For *Rex*1, the control group showed very light markings (Figure 3a), with the 30% LC_50–96 h_-48 h group sharing this appearance (Figure 3b). However, after 72 h at this concentration, a small increase in the number of markings was noticeable (Figure 3c). However, individuals exposed to 50% LC_50–96 h_-C2-48 h showed a considerable difference in the distribution and markings intensity (Figure 3d) when compared to the control and the 30% LC_50–96 h_ groups. This increase continued with the 72 h and 96 h exposures at the 50% LC_50–96 h_ (Figure 3e,f).

For *Rex*6, there was no amplification pattern even at the highest concentration of the pesticide (Figure 4a–f).

## 4. Discussion

When studying repeated sequences in the chromosomes of teleosts to gain a better understanding of transposable elements organization and their role in the diversification of teleost genomes, cytogenetic analysis found a diversity of distributional and locational patterns. Mapping the retrotransposons *Rex*3 and *Rex*6 in *C. macropomum*, *Piaractus mesopotamicus*, and in “tambacu” hybrid (*C. macropomum* X *P. mesopotamicus*), Ribeiro et al., 2017 [41] observed that there were very few markings in *C. macropomum* for *Rex*3, while for *Rex*6, more intense and dispersed markings were presented on several chromosomes. Here, individuals kept at 22 °C, a condition known to be stressful for *C. macropomum*, had shown chromosomes with a high number of *Rex*1 and *Rex*6 site markings, as well as noticeable blocks in the centromeric and telomeric regions of several chromosomal pairs, indicating a heterochromatinization process.

Ferreira et al., 2019 [35] analyzed the distribution of *Rex*1, *Rex*3, and *Rex*6 retroelements and the detection of heterochromatin in *C. macropomum* chromosomes, under different climate change scenarios, proposed by the Intergovernmental Panel on Climate Change (IPCC). The study found heterochromatinization in the genome of individuals exposed to an A2-like scenario (which simulates the most extreme conditions, with an increase of 4 °C and 800 ppm of CO) and this was associated with an increase in the number of *Rex*3 sites.

In the current study, *Rex*3 retroelement sequences proved to be more sensitive than *Rex*1 and *Rex*6 sequences to the chemical stress induced by Trichlorfon, responding to this stressor with activation, possibly due to the disruption of the inactivation control system. However, *Rex*6 was more active in low temperatures, according to Ribeiro et al. (2017) [41]. The activation, transposition, and insertion of these sequences in the genome of *C. macropomum* may be harmful to the individuals involved, since the presence of TEs can deregulate gene and enzymatic machinery. On the other hand, this same mechanism can produce genetic variants and adaptive phenotypes through stochastic drift, and so can be considered a genetic innovator [41,42,43,44]. The association of heterochromatic blocks with *Rex*3 sequences is likely to be mediated by adaptive epigenetic mechanisms that probably aim to suppress the mobilization of these retroelements. This activation and transposition can have transformative effects on the genome of the host, acting either as an adaptive element in stressful environments (whose appearance can help increase genetic variability) or as a selfish parasitic agent within the genome. In addition, based on the increased number of metaphases seen in the most extreme versions of the experiment, Trichlorfon seems to play a substantial role in activating the cell division process.

In summary, following the Trichlorfon exposure, our data revealed that the distribution of *Rex*1 and *Rex*3 retroelements in the *C. macropomum* genome showed a dispersed distribution, but preferentially in terminal regions. This effect was most intense for the treatments involving the longest exposure and the highest concentration. While such markings were quite subtle for *Rex*6, those for the *Rex*3 sites were very marked, as well as the association with heterochromatic blocks, which was visibly prominent. Knowledge about retrotransposition activated by chemical stress remains incomplete and additional studies are essential to generate more information on the distribution and regulation of these sequences. It is also important to improve the experimental design of the processes and mechanisms that involve the high level of functional plasticity that distinguishes vertebrate genomes.

## 5. Conclusions

The retroelements *Rex*1 and *Rex*3 showed a dispersed distribution profile in the karyotypes, whose labeling intensity increased with the exposure time to Trichlorfon. For *Rex*3, this increase was noticeable, and markings were observed at higher concentrations and longer experimentation times. Therefore, we conclude that the stressor agent Trichlorfon, widely used by the pisciculture to mitigate the effects of parasitic infections in fish, has a considerable effect on the mobilization of repetitive sequences, more precisely of *Rex*3 retroelements in the tambaqui genome. Furthermore, Trichlorfon seems to play a substantial role in the cell division process due to the increase in the metaphases number, visualized in the most extreme conditions of the experiment.

## Figures and Tables

**Figure 1 animals-12-01945-f001:**
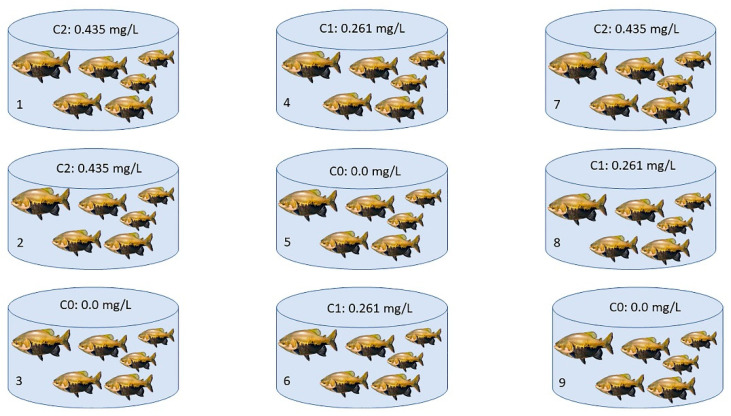
Experimental design using *C. macropomum* as a model organism. The tanks were randomly distributed: the tanks 5, 3, and 9 with C0 corresponding to 0.0 mg/L of Trichlorfon diluted in the water (three tanks with six specimens = 18 specimens analyzed); the tanks 4, 6, and 8 with C1 corresponding to 0.261 mg/L of Trichlorfon; and the tanks 1, 2, and 7 with C2 corresponding to 0.435 mg/L of Trichlorfon. Then 18 fish per treatment were analyzed, six per tank. N sampled—54 fish.

**Figure 2 animals-12-01945-f002:**
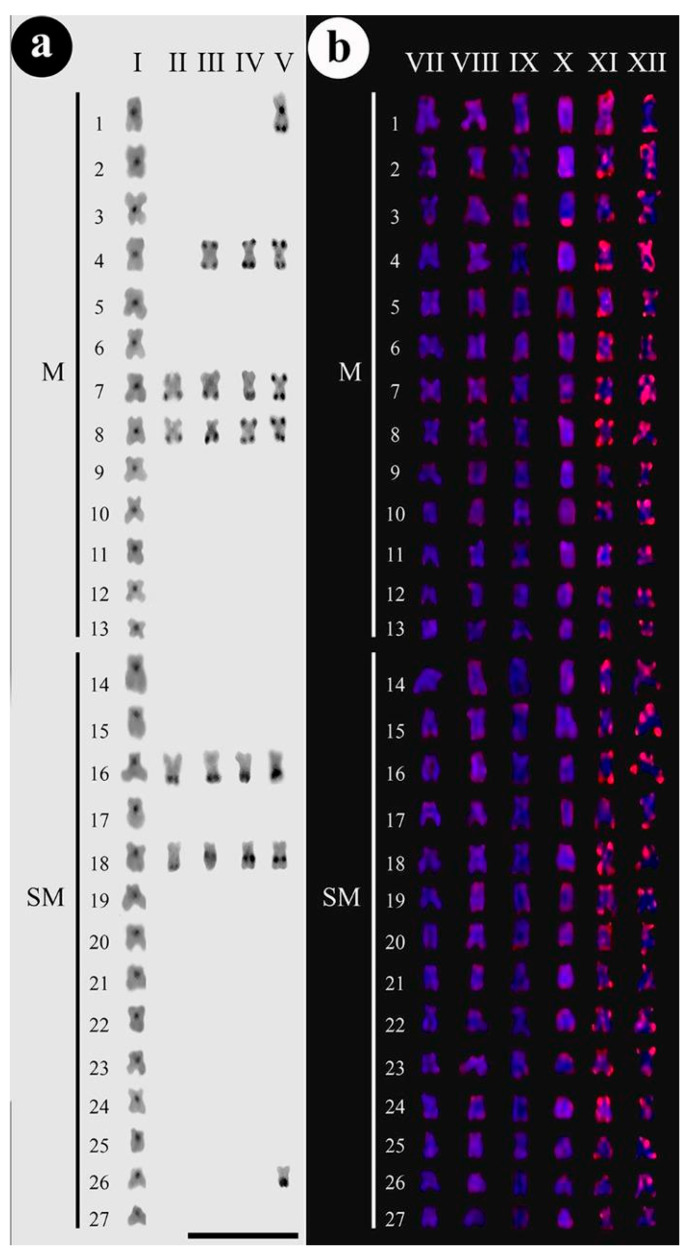
(**a**)-I–C-banding of *C. macropomum* in the control group (C0-0.0 mg/L of Trichlorfon). Heterochromatic regions are marked in the pericentromeric portions of chromosomes, conventional configuration for C-banding of tambaqui. For details, see references [31,32,33,34,35,36,37,38,39,40,41,42,43]. 2(**a**)-II–C-banding after exposition to trichlorfon to 30% LC_50–96 h_-48 h (C1 = 0.261 mg/L). 7, 8, 16, and 18 chromosomes are marked. 2(**a**)-III–C-banding after exposition to 50% LC_50–96 h_-48 h (C2 = 0.435 mg/L), bi-telomeric markings are observed in 4 chromosomes, beyond 7, 8, 16, and 18 chromosomes are marked. 2(**a**)-IV-C-banding after exposition to 50% LC_50–96 h_-72 h, bi-telomeric markings are observed in 4 and 8 chromosomes, terminal markings in 7 and 16 chromosomes, and interstitial markings in 18 chromosomes. 2(**a**)-V–C-banding after exposition to 50% LC_50–96 h_-96 h, bi-telomeric markings are observed in 4, 7, and 8 chromosomes, terminal markings in 16 and 26 chromosomes, centromeric and terminal markings in 1 chromosomes, and interstitial mark in 18 chromosomes. 2(**b**)-VII–FISH of *Rex*3 of *C. macropomum* in control group (C0 = 0.0 mg/L of Trichlorfon). 2(**b**)-VIII-FISH after exposition to 30% LC_50–96 h_-48 h (C1 = 0.261 mg/L of Trichlorfon). Slight markings are observed on the telomeric portions on most chromosomes. The same situation for 2(**b**)-IX and 2(**b**)-X that correspond to 30% LC_50–96 h_-72 h and 50% LC_50–96 h_-48 h of Trichlorfon, respectively. In 2(**b**)-XI and 2(**b**)-XII, that correspond to 50% LC_50–96 h_-72 h and 50% LC_50–96 h_-96 h (extreme condition), intense markings are observed on the telomeres or centromeric regions on most chromosomes. For the most extreme conditions of the experiment, gradual amplification of *Rex*3 markings is observed with the increase in the parasiticide Trichlorfon.

**Figure 3 animals-12-01945-f003:**
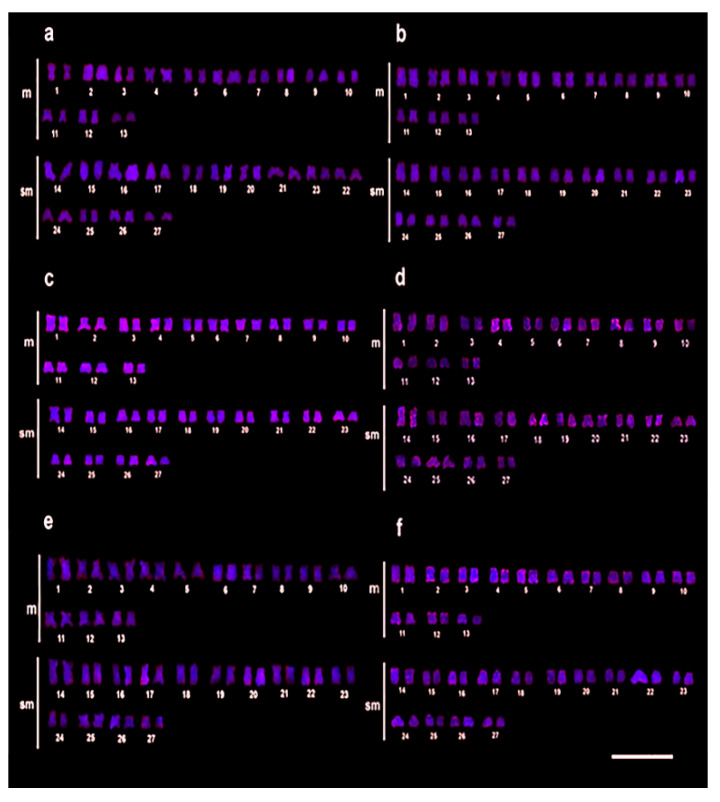
Chromosomal mapping of the *Rex*1 by FISH in *C. macropomum.* (**a**) FISH in the control group (C0 = 0.0 mg/L of Trichlorfon). (**b**) 30% LC_50–96 h_-48 h of Trichlorfon (C1 = 0.261 mg/L). Even at higher concentrations, (**c**) 30% LC_50–96 h_-72 h, (**d**) 50% LC_50–96 h_-48 h (C2 = 0.435 mg/L of Trichlorfon), (**e**) 50% LC_50–96 h_-72 h, (**f**) 50% LC_50–96 h_-96 h, an increase of markings was observed in *Rex*1 retroelements after treatment.

**Figure 4 animals-12-01945-f004:**
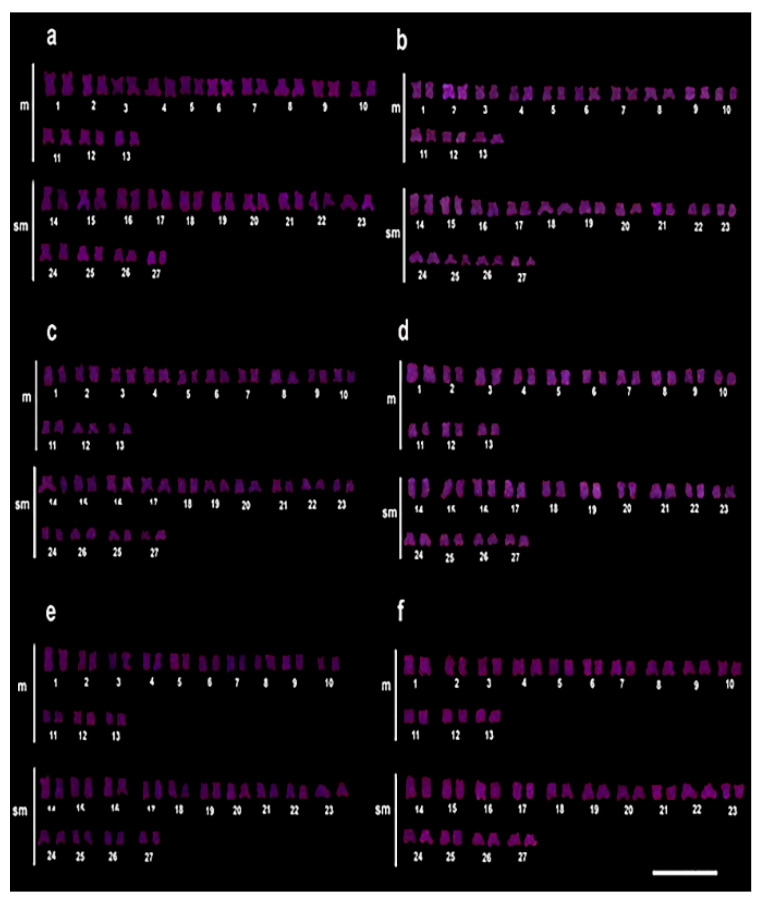
Chromosomal mapping of the *Rex*6 by FISH in *C. macropomum.* (**a**) FISH in the control group (C0 = 0.0 mg/L of Trichlorfon), (**b**) 30% LC_50–96 h_-48 h of Trichlorfon (C1 = 0.261 mg/L), (**c**) 30% LC_50–96 h_-72 h, (**d**) 50% LC_50-96 h_-48 h (C2 = 0.435 mg/L of Trichlorfon), (**e**) 50% LC_50–96 h_-72 h, (**f**) 50% LC_50–96 h_-96 h. It was observed that there was no amplification pattern for *Rex*6 even at the highest concentration of pesticide.

## Data Availability

The data presented in this study are available on request from the corresponding author.

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
