# Peer review of "A Perspective of Molecular Cytogenomics, Toxicology, and Epigenetics for the Increase of Heterochromatic Regions and Retrotransposable Elements in Tambaqui (Colossoma macropomum) Exposed to the Parasiticide Trichlorfon"

_animals, 2022, doi:10.3390/ani12151945_

Round 1

Reviewer 1 Report

In the manuscript “A perspective of molecular cytogenomics toxicology and epigenetics for the increase of heterochromatic regions and retrotransposable elements in tambaqui (Colossoma macropomum) exposed to the parasiticide trichlorfon” the authors aim to evaluate the effects of the antiparasitic Trichlorfon widely used in aquaculture, on the mobilization of repetitive sequences and in particular on the activation of the TEs Rex1, Rex3, Rex6. Tambaqui specimens were submitted to different Trichlorfon concentrations under experimental conditions. The heterochromatin profile showed that these retrotransposons are in the  heterochromatic portions of the chromosomes. It was observed that Trichlorfon has an activation mechanism for these retroelements, especially Rex3, showing the greatest increase in FISH markings when compared to the others. Moreover it has been pointed out an enhanced chromosomal heterochromatinization, possibly mediated by epigenetic adaptive mechanisms.

The topic of the manuscript is important and of general interest, as demonstrated by the increasing number of articles reporting a link between the genomic abundance of TEs and the adaptation to specific environmental condition in plants, and in animals and in particular in fish [Carotti et al, 2021)]. In fish Rex3 is the most diffused TE (Volf et al., 1999; Ferreira et al., 2011) and differences of the distribution/concentration of Rex3 are known (Cioffi et al, 2010). 

In my opinion, the principal topic of this research, to demonstrate physically on the chromosomes an increase in the amount of Rex3  after a treatment with a stressing chemical, is very interesting because it highlights which chromosome regions are interested in the mobilization of the TE. Based on the experimental evidence of this work, the telomeric region appears to be the one where these sequences are located and the apparent expansion of telomeric heterochromatic regions deserves to be investigated. The presence of expanded telomeres in many species of animals and plants is well documented and the possible relation to the activation/regulation of TEs is very intriguing.  As a suggestion for a future work, a contemporary FISH of Rex3 and telomeric sequences could be very interesting.

Moreover this manuscript highlights the mitogen role of the parasiticide trichlorfon, and this finding could be very useful in fish cell culture.

However, the enhanced chromosomal heterochromatinization must be demonstrated with certainty, keeping in mind the interindividual intrinsic variability of the heterochromatic regions. To do this in my opinion this manuscript lacks the following, informative, experimental data:

1)      the authors should clearly report the experimental protocol: i) the numbers of individuals studied for each step, ii) in how many individuals they obtained the showed results after the different steps and if the results were the same in all the individuals studied. Moreover iii) if in the control sample interindividual variability regarding the heterochromatin distribution/size was present

2)      how the chromosomes have been obtained? By cell culture o by a direct method? In this research the use of cell colture could allow to study in the same individuals the heterochromatin/Tes distribution before and after the treatments with the antiparasitic compound and this could exclude at all the doubt about the interindividual variability of heterochromatic regions.

Moreover, from what I could see, the legends of the figures are missing and in the Results section the results regarding the Rex3 marking distribution are sometimes referred as present in Fig 2 (line 186,194) and sometimes in Fig. 3 (Lines 187, 194)

Line 285 I suggest: seems to play a mitogen role, activating the cell division process

Author Response

  • The authors should clearly report the experimental protocol: i) the numbers of individuals studied for each step, ii) in how many individuals they obtained the showed results after the different steps and if the results were the same in all the individuals studied. Moreover iii) if in the control sample interindividual variability regarding the heterochromatin distribution/size was present

Answer: i) we analyzed 18 specimens per treatment, and 60 metaphases per treatment. ii) we did not observe polymorphism in the individuals analyzed. iii) according to Figure 2a-I, we did not observe interindividual variation in relation to the heterochromatin pattern.

  • how the chromosomes have been obtained? By cell culture o by a direct method? In this research the use of cell colture could allow to study in the same individuals the heterochromatin/Tes distribution before and after the treatments with the antiparasitic compound and this could exclude at all the doubt about the interindividual variability of heterochromatic regions.

Answer: According to the protocol by Gold et al. (1990), which used cellular culture.

Moreover, from what I could see, the legends of the figures are missing and in the Results section the results regarding the Rex3 marking distribution are sometimes referred as present in Fig 2 (line 186,194) and sometimes in Fig. 3 (Lines 187, 194).

Answer: correction made.

Line 285 I suggest: seems to play a mitogen role, activating the cell division process

Answer: suggestion accepted.

Reviewer 2 Report

e manuscript animals-1681056 titled: A perspective of molecular cytogenomics toxicology and epigenetics for the increase of heterochromatic regions and re-trotransposable elements in tambaqui (Colossoma macropomum) exposed to the parasiticide trichlorfon

Is about re-trotransposable (TE) mapping by Fluorescent in situ hybridization in Tolostei fish genome after treatment with pesticide in aquaculture; authors claims that after treatment with different concentration of parasiticide trichlorfon there is an incrementing of transposition elements and heterochromatin.

The manuscript shows new evidences worthy to be published even if many points have to be better clarified in the main parts of the manuscript:

see PDF

Author Response

 The manuscript animals-1681056 titled: A perspective of molecular cytogenomics toxicology and epigenetics for the increase of heterochromatic regions and re-trotransposable elements in tambaqui (Colossoma macropomum) exposed to the parasiticide trichlorfon

Is about re-trotransposable (TE) mapping by Fluorescent in situ hybridization in Tolostei fish genome after treatment with pesticide in aquaculture; authors claims that after treatment with different concentration of parasiticide trichlorfon there is an incrementing of transposition elements and heterochromatin.

The manuscript shows new evidences worthy to be published even if many points have to be better clarified in the main parts of the manuscript:

Introduction

Introduction have to been extended regarding the approach used.

Since exist many approaches to find DNA sequences and in this article the approach used is the molecular cytogenetics ones, it is necessary introduce it. Thus you should speak about Classic and molecular cytogenomic and in particular of the FISH to maps DNA sequences probes into chromosomes and the many references linked to it as for example:

1) Dumas, F.; Sineo, L. The evolution of human synteny 4 by mapping sub-chromosomal specific probes in Primates. Caryologia 517 2014, 67, 281–291 5

2) Dumas, F.; Sineo, L.; Ishida, T. Taxonomic identifcation of Aotus (Platyrrhinae) through cytogenetics. J. Biol. Res 2015, 88, 65- 519

Furthermore, in particular I would speak about TE mapping in other groups available as for example:

1) Ceraulo, S., Perelman, P. L., Dumas, F. 2021b. Massive LINE-1 retrotransposon enrichment in tamarins of the Cebidae family (Platyrrhini, Primates) and its significance for genome evolution. Journal of Zoological Systematics and Evolutionary Research.DOI: 10.1111/jzs.12536

2) Ceraulo S, Milioto V, Dumas F (2021c) Centromeric enrichment of LINE-1 retrotransposon in two species of South American monkeys Alouatta belzebul and Ateles nancymaae (Platyrrhini, Primates). Caryologia 74(4): 111-119. doi: 10.36253/caryologia-1296

And of course in fish and toleostei in particular [20-27, 31]

Also seems to be important to define in the introduction what constitutional heterochromatin is

Answer: we insert a short paragraph talking about cytogenetic mapping by FISH, added a brief definition of heterochromatin, and cited some papers about Rex elements.

Material and methods

Material and methods section need a big revision

suggest to better explain experiment 1: how many samples are in the control group and how many with the different concentration; this point is important in order to understand and clearing if you have evaluated the presence of normal specie specific polymorphisms due for example to rearrangements and if the difference in TE amplification you find are really due to the increase of pesticide.

Answer: We analyzed 54 fish. There were 18 fish per treatment at concentrations C0, C1, and C2 or 6 fish per tank. At least 60 metaphases were analyzed for each condition of the experiment. We did not verify C-band polymorphism in the control group. The distribution pattern of the elements Rex 1, 3, and 6 and C-band for C. macropomum have been described in other studies.

I see in the picture different concentration in respect with what explained in the text; better explain picture 1 both in the text and in legends.

Report cytogenetic methods and their references: cell cultures, harvesting of mitotic chromosomes, C banding methods….

Answer: suggestion accepted.

Results

Results need to better organized too

Results are not so well defined. I see in the pictures the TE amplification after treatment especially of rex 1and rex 3 while no clear result for rex 6. For the first two TE I also suggest to do not make such a big differences regarding level of amplification even because it is not possible to really measure them quantitatively. I would say that there is a gradual amplification of TE signals with the incrementing of the concentration of parasiticide trichlorfon.

Also I’m not agree with Signals over the entire Chromosomes for rex 1 and 3; they are indeed very amplified signals at telomeres and in a few centromeric position in all samples

I would say that there are not Rex6 amplification pattern even at high concentration of pesticide.

Control figures order and their introduction in the text, especially for figure 2; invert figure order… put legends in all picture to better explain them

Answer: suggestion accepted.

Report previous data on C bands in these animals for example: Ferreira et al 2019

Answer: suggestion accepted.

C bands obtained in the present work are just for animals analyzed to detect Rex 1? Please better explain the point

Answer: We analyzed the C-Band and verified that there was a correlation with the distribution pattern of the Rex3 element. For the other retroelements, we did not achieve a direct correlation.

Discussion

Discussion need to be rewrite

I would adjust discussion and conclusion according a more cautioned analysis of results. Better explain which are the current results especially comparing them with previously data published (Ferreira et al 2019) on other samples and condition since the results on rex TE here find are similar to that find in Ferreira. Making clear inferences about possible epigenetic and adaptive scenario starting from your data. Avoid phrases repetition

Answer: suggestion accepted.

References

Please check all references

for example, the [32 and 33] are missing in the text and explain please the methods.

In the first sentences in the discussion are cited [30-38] in regard to epigenetic factors, but for example citation 32 to 35 are not inherent

Answer: suggestion accepted.

Reviewer 3 Report

In the manuscript “A perspective of molecular cytogenomics toxicology and epigenetics for the increase of heterochromatic regions and retrotransposable elements in tambaqui (Colossoma macropomum) exposed to the parasiticide trichlorfon” the authors aimed to evaluate the effects of the antiparasitic Trichlorfon in the tambaqui genome. They used the FISH technique and the heterochromatin standard C-band technique.

Organochlorophosphates, such as trichlorfon, are the class of pesticides most commonly used by those involved in commercial pisciculture. The environment scientific evidence has revealed their toxic effects onto animal and, as a consequence, on human health. These topics are of great relevance and this manuscript could provide important reference information that are of considerable interest.

The manuscript is well written and the methodology appears correct.

 Specific comments

11. I recommend to improving figure 1 or introducing a graphical abstract to better illustrate the experimental design and the aims of the study.

22. All the explanation of figures are missing

Author Response

The manuscript is well written and the methodology appears correct.

 Specific comments

  1. I recommend to improving figure 1 or introducing a graphical abstract to better illustrate the experimental design and the aims of the study.

Answer: suggestion accepted.

  1. All the explanation of figures are missing

Answer: correction made.

Reviewer 4 Report

Dear Dr. da Silva,

I have carefully read your manuscript entitled "A perspective of molecular cytogenomics toxicology and epigenetics for the increase of heterochromatic regions and retrotransposable elements in tambaqui (Colossoma macropomum) exposed to the parasiticide trichlorfon". I believe it contains new important information on the effect of Trichlorfon on the distribution of heterochromatic segments and retrotransposable elements in this fish species, and therefore this manuscript could be published in Animals. However, your figures miss separate descriptions, and therefore figure legends should be inlcuded; please also change "Figure 3b-VII" to "Figure 2b-VII" (line 187 of the manuscript). Moreover, I suggest a thorough revision of the paper's language, preferably by a native English speaker. You apparently meant "anti-digoxigenin-rhodamine" and "Fluorescent" instead of "a rhodamine anti-di" and "Flourescentgoxigenin" (lines 130-131). Analogously, did you mean "duration" instead of just "du" (line 168)? Furthermore, please change "Digoxigenine" to "Digoxigenin" (line 128). In addition, spelling of the chemical's name (Trichlorfon) on the left and right of Figure 1 should be carefully checked. In the References section, all genus and species names must be given in italics (see lines 332, 350-351, 354, 366, 383, 392, and 423).

Author Response

I have carefully read your manuscript entitled "A perspective of molecular cytogenomics toxicology and epigenetics for the increase of heterochromatic regions and retrotransposable elements in tambaqui (Colossoma macropomum) exposed to the parasiticide trichlorfon". I believe it contains new important information on the effect of Trichlorfon on the distribution of heterochromatic segments and retrotransposable elements in this fish species, and therefore this manuscript could be published in Animals. However, your figures miss separate descriptions, and therefore figure legends should be inlcuded; please also change "Figure 3b-VII" to "Figure 2b-VII" (line 187 of the manuscript). Moreover, I suggest a thorough revision of the paper's language, preferably by a native English speaker. You apparently meant "anti-digoxigenin-rhodamine" and "Fluorescent" instead of "a rhodamine anti-di" and "Flourescentgoxigenin" (lines 130-131). Analogously, did you mean "duration" instead of just "du" (line 168)? Furthermore, please change "Digoxigenine" to "Digoxigenin" (line 128). 

Answer: correction made. The paper was previously translated by a native English speaker.

In addition, spelling of the chemical's name (Trichlorfon) on the left and right of Figure 1 should be carefully checked. In the References section, all genus and species names must be given in italics (see lines 332, 350-351, 354, 366, 383, 392, and 423).

Answer: suggestion accepted.

Round 2

Reviewer 1 Report

I think that the authors should insert their answers in the manuscript,  in the materials and methods section as regards the number of specimens studied  in each experimental step and in the result section as regards the absence of polymorphism in the studied individuals

Author Response

All suggestions was accepted.

Best regards,

Hallana.

Reviewer 2 Report

The article (ID1681056)  sound better ofther changes suggested. However, before publication would need minor revision:

Methods

Still experiment need clarifications:

Text in lines 109-12 and 117-119 in respect to figure 1

Also better explain figure 1 in legends (each groups in the three experiment have C1, C1, C2 or it random) including timing in agreement with text.

Results

194-206 Can be resumed since the same points are explained in legends Fig 2aI. It would be more fluent.

Line 207-208 Delete because-no diffuse signals

220-228 adjust text in regard to Fig 2b VIII because no diffuse signals are at 30% concentration but instead slight signals at telomeric ends on the majority of chromosomes

Fig 2b XII very bright signals telomeric ends on the majority of chromosomes or at centromeric position (presumably linked with the amplification of the sequences after treatments)

Explain better relation with C bands; indeed, the increase of FISH signals is both on chromosomes with evident new C bands (after treatment) but also on chromosomes without evident new C bands (after treatment) …

247-248 starting form…and even at higher concentration

301- delete Rex 6 (no signals)

312 delete Rex 6

Would be worthy to say something about REX 3 not functional in Rebeiro 2017 and functional in others context? The same for Rex 6 no functional here but functional in other Context?

Author Response

(The authors gave the same response as above.)
